# Stable Isotope Analysis Reveals Habitat-Driven Dietary Niches of *Lepus europaeus*

**DOI:** 10.3390/ani16010015

**Published:** 2025-12-20

**Authors:** Linas Balčiauskas, Rasa Vaitkevičiūtė-Koklevičienė, Andrius Garbaras, Jolanta Stankevičiūtė, Inga Garbarienė, Laima Balčiauskienė

**Affiliations:** 1State Scientific Research Institute Nature Research Centre, LT-08412 Vilnius, Lithuania; laima.balciauskiene@gamtc.lt; 2Agriculture Academy, Vytautas Magnus University, Studentų 11, Akademija, LT-53361 Kaunas, Lithuania; rasa.vaitkeviciute1@vdu.lt (R.V.-K.); jolanta.stankeviciute1@vdu.lt (J.S.); 3VšĮ Forest 4.0, LT-53361 Akademija, Lithuania; 4Center for Physical Sciences and Technology, Saulėtekio Ave. 3, LT-10257 Vilnius, Lithuania; andrius.garbaras@ftmc.lt (A.G.); inga.garbariene@ftmc.lt (I.G.)

**Keywords:** European brown hare, nitrogen and carbon isotopes, diet proxy, Lithuania, Poland

## Abstract

The European brown hare (*Lepus europaeus*) is a common inhabitant of open fields and farmlands. However, its population has declined across Europe due to changes in agriculture. Modern large-scale farming has reduced the variety of plants and habitats on which hares depend for food and shelter. In this study, we examined the diets of hares in Lithuania and Poland by analyzing stable isotopes in their hair. Differences in carbon and nitrogen values can serve as a proxy for hare diets. We found that hares living in Lithuania’s varied landscapes had a broader diet than those in Poland’s orchard areas, which had a more limited range of foods. Male and female hares, as well as young and old ones, had similar diets. Our results indicate that diverse farmland with different crops, grassy areas, and shrubs provides better feeding opportunities for hares. Promoting such landscape diversity can support healthy hare populations and more balanced farmland ecosystems.

## 1. Introduction

### 1.1. Population Decline and Environmental Drivers of the European Brown Hare

The European brown hare (*Lepus europaeus* Pallas, 1778) originated in the open steppe grasslands of Eurasia and has adapted very successfully to mixed arable agricultural environments. However, declines in *L. europaeus* populations have been reported across many European countries in recent decades of the 20th century [1]. In these landscapes, the dominance of arable fields and intensive land-use practices, characterized by limited crop and habitat diversity and intensive mechanization, appears to be the primary driver of population declines in numerous farmland species [2]. Low *L. europaeus* densities are generally attributed to these modern agricultural practices, whereas higher densities tend to occur in regions with mild climates, nutrient-rich soils, and varied vegetation [3]. Scattered woody vegetation, represented by transitional woodland-shrub and small woody features, and warmer winter temperatures have been identified as key predictors of increased *L. europaeus* density. This suggests that promoting near-natural landscape heterogeneity through incentives and regulations can significantly improve habitat quality and support this popular mammal’s population [4].

Unlike rabbits, *L. europaeus* does not burrow and gives birth to precocial young. This species is well-adapted to agricultural landscapes and prefers open grasslands and pastures bordered by hedgerows and bushes. Despite significant population declines in some regions, the species remains classified as “Least Concern” by the IUCN [5].

Since the 1960s, *L. europaeus* populations have declined across Europe, with early reductions possibly linked to misidentified paraquat poisoning or emerging diseases such as European Brown Hare Syndrome. However, long-term declines are primarily attributed to changes in farmland management, such as reduced crop and landscape diversity, which affect hare nutrition. Although predation and disease contribute to mortality, they are not considered primary drivers [6].

The *L. europaeus* population in Lithuania has experienced significant long-term fluctuations, with peak harvests in the 1950s and 1960s, followed by a steady decline. The average annual harvest was about 9500 individuals from 2000 to 2005, though there were pronounced regional disparities [7]. In Poland, *L. europaeus* populations have declined significantly since the late 1970s, with the most pronounced reductions occurring in the 1990s [8]. A long-term study (1965–2018) in central Poland documented a decline in density from over 30 to just 1–2 individuals per 100 hectares (ha), with recent autumn densities matching spring levels. This indicates low recruitment and no density advantage in fields over forests, likely due to the declining quality of arable land. Additionally, hares currently comprise only 0.1% of the red fox diet, down from 13% historically [1].

Key factors influencing the abundance of *L. europaeus* in Lithuania include climatic conditions, hunting intensity, predation (particularly by red foxes), and parasitic infections. Spring and summer climatic patterns, such as warm and dry weather, strongly correlate with higher population levels [7]. In Poland, *L. europaeus* abundance and decline (2000–2014) were negatively correlated with increasing farm size, forest cover, and the proportion of grassland and cereal crops. However, no relationship was found with red fox density, suggesting that habitat transformation due to agricultural intensification was the main cause [9]. Additional studies confirm that field enlargement, crop homogenization, and loss of unmanaged vegetation reduce food quality and availability for hares [10]. A comparison of two regions with similar ecological conditions but differing agricultural intensities revealed population increases only in the less intensively farmed area. This finding reinforces the critical role of farming practices in *L. europaeus* declines [1].

### 1.2. European Hare Diet

It is important to assess the nutritional needs of *L. europaeus* for effective habitat management, especially in agricultural landscapes where this herbivorous species is declining [11]. They practice caecotrophy, a specialized digestion process allowing nutrient recovery, which drives their preference for high-quality, low-fiber foods rich in fats and proteins. Its diet consists of grasses, various weeds (e.g., Asteraceae and Fabaceae), and crops such as wheat, barley, oats, and rye; however, wild plants are generally preferred when available [5]. In winter, *L. europaeus* diet shifts to woody material, and they may consume supplemental feed provided by hunters. Monocultures and reduced crop diversity negatively impact nutrition and populations. However, agri-environmental practices like crop rotation, polycultures, and wildflower strips can improve habitat quality and food availability [11].

In Lithuania’s pure pine forests, the winter diet of *L. europaeus* consists predominantly of woody plant shoots. Preferred species include Scots pine (*Pinus sylvestris*), birch (*Betula pendula*), rowan (*Sorbus aucuparia*), alder buckthorn (*Frangula alnus*), and dwarf shrub bilberry (*Vaccinium myrtillus*). Shoots of *Sarothamnus scoparius* and *Vaccinium myrtillus* were particularly heavily browsed, especially under low snow cover conditions [12]. There were no other investigations on the *L. europaeus* diet in Lithuania, Latvia, or Estonia.

The diet of *L. europaeus* in Central Europe consists of many plant species, though two to three species account for about half of its intake [13]. Agricultural intensification has reduced wild plant diversity, creating nutritional gaps in the summer after winter crop harvests that hinder reproduction. Mitigation measures include green corridors, organic farming, predator control, and hare farming to support population growth [14].

In extreme environments such as Mount Vesuvius in Italy, *L. europaeus* exhibits strong dietary specialization. Over 86% of their diet consists of Fabaceae, primarily *Galega officinalis* and *Lupinus angustifolius*, despite the rarity of these plants in the area [15]. Other locally available wild and cultivated plants were found only in trace amounts. These findings underscore the ecological adaptability and selective feeding behavior of *L. europaeus*, both of which are essential considerations in conservation planning, particularly in the context of post-fire habitat changes that alter trophic availability.

### 1.3. Hare Diet Investigation Methods

Investigating the diet of *L. europaeus* primarily relies on two methods: microhistological analysis of stomach contents and fecal analysis [11]. Microhistological analysis involves identifying plant epidermal fragments under a microscope from stomach contents that have been dissected. This method provides a detailed picture of the animal’s diet, enabling researchers to detect even partially digested food items. This method has been widely used in studies such as those by Sokos et al. [16], who sampled the stomach contents of hares harvested during hunting seasons to assess seasonal and demographic dietary variations.

Fecal microhistological analysis is a non-invasive alternative that is often used in field studies. Katona and Altbäcker [17] emphasized the importance of optimizing this method by determining the minimum sample sizes at various levels (e.g., number of pellets, subsamples, and epidermis fragments per slide) to ensure reliable results. They recommend analyzing at least ten independent droppings and one hundred epidermal fragments per sample to account for high dietary variability among individuals and enhance the accuracy of relative frequency estimates of forage classes.

Additionally, field studies may incorporate pellet group counts and browsing intensity measurements to estimate the feeding pressure of *L. europaeus* on woody vegetation, especially in forested environments [12]. These methods complement microscopic analysis by providing information on spatial feeding patterns and habitat use during non-vegetative periods. Overall, integrating direct and indirect methods provides a comprehensive understanding of the hare’s diet and its ecological implications.

A novel method for studying diet involves DNA metabarcoding and high-throughput sequencing of DNA extracted from fecal pellets. This approach enables precise identification of ingested plant species. Before dietary analysis, the fecal DNA was verified as originating from *L. europaeus* using high-resolution melting analysis [15].

Although these methods are effective, they are labor-intensive and expensive. Therefore, we are looking for an easier, faster method. Stable isotope analysis of keratinous tissues, such as hair, is a well-established technique for determining the diets of wild mammals by tracking the carbon (δ^13^C) and nitrogen (δ^15^N) isotopic signatures [18]. Stable isotopes record nutrients that contribute to tissue growth, making them effective proxies for reconstructing diet over time [19]. Since different tissues incorporate isotopes at different rates, they can reflect dietary intake over periods ranging from weeks to decades, depending on the tissue analyzed [20].

Carbon (δ^13^C) and nitrogen (δ^15^N) isotopes provide complementary dietary information in herbivores. δ^13^C reflects the types of plants assimilated and differences among habitats or agricultural resources. Meanwhile, δ^15^N indicates trophic position and protein sources and often increases with the consumption of nitrogen-rich or fertilized plants. Thus, stable isotope ratios in hair provide an averaged representation of diet and habitat use over time [18,19,20]. The method was used for dietary studies of wild mammals with applications across various European species, including wild boars [21], brown bears [22], and small mammals [23]. The analysis of δ^15^N and δ^13^C values in the hair of 50 mammal species in the Białowieża Forest in Poland over six decades (1946–2011) revealed a significant decrease in δ^15^N, particularly among herbivores, likely due to nitrogen deposition. Additionally, a decline in δ^13^C was observed in bats, suggesting changes in the food web. These findings highlight the importance of mammal communities as indicators of environmental change [24]. However, for *L. europaeus*, isotope data from 1950 to 2009 included just seven individuals, and show marked temporal variation, indicating substantial shifts in trophic sources over time and supporting the species’ sensitivity to environmental and agricultural change.

Although the analysis of stable isotopes in hair is a well-established method for studying the diets of mammals, it has been applied only marginally to assess the diets of *L. europaeus*. This omission represents a gap in the literature and an opportunity for future research.

The aim of this investigation was to assess the diet of *L. europaeus* in Lithuania using a stable isotope-based evaluation of the trophic niche in different habitats, and to compare the results to those from Poland. Because the country identity in our study reflects the sampling design rather than true nationwide variation, with Lithuania encompassing several habitat types and Poland comprising only a single orchard site, these differences largely reflect habitat scope rather than geopolitical boundaries. Additionally, we tested for differences in the *L. europaeus* diet between males and females and between different age groups. By analyzing individuals from contrasting environments, we provide a contemporary baseline for the species’ isotopic niche within modern agricultural landscapes.

## 2. Materials and Methods

### 2.1. Study Sites and Habitats

Hair was collected from animals, legally hunted in Lithuania and Poland. In Lithuania, 45 sites were covered with a number of *L. europaeus* from 1 to 8 in each, while in Poland, all 68 sampled individuals originated from the same site, the Mogielnica apple orchards (Figure 1).

Habitats in Lithuania covered (agricultural) fields, forests, riverside, shrubs, and settlements. In Poland, the only habitat was an orchard (apple trees, currant plantations, and greenery). Grass moving and plowing between rows of apple trees was used in younger orchards. Old orchards were neglected; grass was high and not mown. Surrounding areas are forest, shrub, and fields.

### 2.2. Hair Collection and Sample Size

The hair of *L. europaeus* was collected either directly at the hunting site or later during dissection in the laboratory by clipping a 4–5 mm wide tuft of hair from the shoulders. The collected hair was stored in plastic bags in a freezer at a temperature of −20 °C.

During the 2023/2024 and 2024/2025 hunting seasons, we collected hair from 151 individuals. The sample breakdown is presented in Table 1.

### 2.3. Age Identification in Sampled Hares

After comparing two methods for determining the age of European hares, dry eye lens weight and Stroh’s method, which is based on ulna coalescence [25], we further used the former in Lithuania. This method was also approved in other countries [26]. The method is based on the fact that the weight of the dry eye lens increases with age due to protein accumulation. This method is reliable and independent of external factors, such as season or weather.

To prepare the eye lenses for age determination, they were extracted from hunted animals and fixed in a 10% formalin solution to preserve tissue integrity. The extracted lenses were then dried in an oven at 100 °C for 48 h until they reached a stable mass. The dried lenses were weighed with a precision of 1 mg. The weight of both lenses from the same individual was averaged.

The following simplified age classification based on average length and weight was used: class I: weight < 280 mg, up to 1 year; class II: 280–310 mg, 1–2 years; class III: 310–370 mg, 2–3 years; class IV: >370 mg, over 3 years. Compared to the age classification used in [26], we included all hares younger than 1 year in the same group.

### 2.4. Stable Isotope Analysis of Hair

Approximately 1 mg of hair was required for the analysis of stable nitrogen and carbon isotopes. The sample preparation procedure: hair samples were transferred to test tubes and rinsed three times with deionized water while undergoing sonication to remove surface debris. Then, the samples were immersed in a 2:1 methanol–chloroform solution for 10 min, followed by three additional rinses with deionized water. This process was repeated with a two-hour soaking period to ensure the thorough removal of potential contaminants. Finally, the samples were freeze-dried for 48 h and sealed in tin capsules for stable isotope analysis.

Stable isotope analysis: The stable carbon and nitrogen isotopic compositions were determined using a Thermo Delta V continuous flow isotope ratio mass spectrometer (Thermo, Bremen, Germany) coupled with a Thermo Flash EA 1112 elemental analyzer (Thermo, Bremen, Germany) at the Center for Physical Sciences and Technology in Vilnius, Lithuania. The stable carbon and nitrogen isotopic compositions were calibrated relative to the VPDB and AIR scales using USGS24, IAEA-CH3, IAEA-N1, IAEA-N2, and IAEA-600. The precision (u(Rw)) was determined to be ±0.11‰ for δ^13^C and ±0.14‰ for δ^15^N based on repeated measurements of the calibration standards and sample replicates. The total analytical uncertainty was estimated to be ±0.17‰ for δ^13^C and ±0.19‰ for δ^15^N.

### 2.5. Statistical Data Analyses

Before conducting data analyses, we tested the normality of the *δ*^13^C and *δ*^15^N distribution using the Kolmogorov–Smirnov D test. Both parameters conformed to a normal distribution (D = 0.062, *p* > 0.05 for δ^13^C; D = 0.051, *p* > 0.05 for δ^15^N). Therefore, parametric methods were used further.

We used a general linear model (GLM) for the analysis. We used the values of δ^13^C and δ^15^N as dependent variables and animal age, country, and habitat group as categorical predictors. We used the year and month as continuous predictors. We used Hotelling’s T^2^ for two groups or Wilks’ lambda for several groups to test the significance of the model and eta-squared to test the influence of the categorical factors. We applied Tukey’s HSD test with unequal N for post hoc analysis. The minimum confidence level was set at *p* < 0.05. At the *p* < 0.10 level, we assumed a trend would exist, but not a difference.

We quantified the isotopic niche width and overlap among categories (e.g., countries, habitats, gender, and age) using the SIBER framework [27]. We calculated the standard ellipse area (SEA) to represent the core isotopic niche, which contains approximately 40% of the data points. For small sample sizes, we applied small-sample corrections (SEAc). Additionally, we estimated a Bayesian SEA (SEAb) using a parametric bootstrap approach that resampled from a fitted bivariate normal distribution for each country. Finally, we quantified the overlap in the isotopic niches of countries as the proportion of overlap between their 40% ellipses. We estimated this overlap using a Monte Carlo simulation and expressed it as a percentage of the smaller ellipse area.

Isotopic biplots were drawn in SigmaPlot ver. 12.5 (Systat Software Inc., San Jose, CA, USA), calculations were performed in Statistica for Windows, version 6.0 (StatSoft, Inc., Tulsa, OK, USA).

## 3. Results

### 3.1. Environmental and Temporal Predictors of Stable Isotope Variation

To evaluate the impact of environmental, spatial, and temporal factors on the isotopic composition of *L. europaeus* hair, GLMs were fitted for both δ^15^N and δ^13^C values. The results of these analyses are summarized in Appendix A.

The model accounted for a moderate proportion of the variance in δ^13^C (R^2^ = 0.367; adjusted R^2^ = 0.314). Country, habitat group, and year had significant effects on δ^13^C, while month had a marginal effect, and age was not significant. Regarding δ^15^N variance, the model explained a relatively small proportion (R^2^ = 0.128; adjusted R^2^ = 0.054). Habitat group and month significantly influenced δ^15^N, while age, country, and year were not significant.

### 3.2. Stable Isotope Variation Across Countries

There was a significant difference in average δ^13^C and δ^15^N values in the hair of Lithuanian and Polish *L. europaeus* populations (Hotelling’s T^2^ = 46.9, *p* < 0.0001). Lithuanian *L. europaeus* had significantly lower δ^13^C values (t = 6.83, *p* < 0.0001) than Polish individuals (Table 2). However, δ^15^N values were similar between countries on average (t = 0.65, *p* = 0.52). Lithuanian hares showed greater variance (Figure 2), consistent with broader dietary and habitat diversity.

Differences between the two countries were also evident in terms of isotopic niches (Appendix A). The Lithuanian *L. europaeus* had the widest isotopic niche (SEAb_mean = 3.66), suggesting a broad range of δ^13^C and δ^15^N values and a variety of food resources or habitats (Appendix A). In contrast, the Polish *L. europaeus* had the smallest isotopic niche width (SEAb_mean = 1.84), implying a narrower range of isotopic values. Appendix A shows these regularities with a wider central ellipse for the Lithuanian hares compared to a narrower ellipse for the Polish sample. The overlap between the two populations’ dietary resources is indicated by the overlap between the ellipses, but the Lithuanian hares exploit a much wider trophic space.

### 3.3. Habitat Influence on Isotopic Signatures

The multivariate test revealed significant differences in the combined δ^13^C and δ^15^N values across habitats (Wilks λ = 0.674, *p* = 0.0025). This confirmed that the isotopic composition differs significantly across habitat types and supports the hypothesis of habitat-driven dietary variation in *L. europaeus* (Table 3).

As for δ^13^C, there was significant variation across habitats (F_5,132_ = 10.74, *p* < 0.0001). The most depleted value was found in the settlement (just one individual). The highest δ^13^C values were found in *L. europaeus* from orchards (Figure 3) and significantly exceeded those in other habitats (post hoc, *p* < 0.05); they were closest to the values in fields. δ^15^N variation across habitats was not significant (F_5,132_ = 1.76, *p* = 0.12), with the lowest values found in riversides.

### 3.4. Temporal Variation in Isotopic Signatures

There were no significant differences in the stable isotope concentrations of *L. europaeus* hair between the years of the study (δ^13^C: F_2,148_ = 1.80, *p* = 0.16; δ^15^N: F = 0.39, *p* = 0.67). Variation in both countries was small (see Table 4), and isotopic niches indicated full overlap (see Appendix A).

In general, the monthly variability of δ^13^C was significant (F = 14.00, *p* < 0.0001), while the trend of δ^15^N indicated a trend (F = 2.99, *p* = 0.053). However, these differences were not significant in Lithuania (Table 4). In Poland, the concentration of δ^13^C increased in December compared to November (post hoc, *p* = 0.003), while the concentration of δ^15^N decreased (*p* < 0.01). Monthly isotopic niches showed significant overlap (Appendix A).

### 3.5. Sex and Age Class Differences in Isotopic Signatures

No differences in δ^13^C or δ^15^N values were observed in *L. europaeus* with respect to sex or age (Figure 4). The statistics for gender (δ^13^C: F_1,63_ = 0.07, *p* = 0.80; δ^15^N: F = 0.30, *p* = 0.59) and age (δ^13^C: F_3,139_ = 0.36, *p* = 0.78; δ^15^N: F = 0.43, *p* = 0.73) were not significant. The mean isotopic values were similar between males and females, as well as among all age classes within each country (Appendix A).

## 4. Discussion

### 4.1. Decrease in European Brown Hare Numbers

Lithuanian long-term survey and hunting data are taken from national game statistics and summarized in [28]; Polish data were extracted from [9]. The population of *L. europaeus* in Lithuania exhibited a clear long-term fluctuation pattern (Figure 5a). From the mid-1930s to the early 1950s, the population increased steadily. Then, there was a peak in the mid-1960s, when the population reached nearly 300,000. After this peak, numbers declined sharply through the 1970s and continued decreasing into the 1980s and early 1990s until reaching a low point. There was a modest recovery in the late 1990s, but the population plateaued afterward. Surveys ceased in the early 2000s [28].

Hunting records for *L. europaeus* in Lithuania, dating back to the 1950s, show a long-term decline that mirrors the population trend observed in the survey (Figure 5a). Annual harvests exceeded 50,000 in the early decades, but they dropped sharply after the late 1960s. Harvest levels continued to decrease throughout the 1980s and fell below 10,000 by the 2000s. Though a modest increase is evident from 2015 to 2024, harvest levels remain far below historical highs. Overall, the pattern indicates reduced hunting pressure in response to declining population densities due to regulatory constraints.

Similar tendencies were observed in both countries, namely a decrease in the hunting bag (Figure 5b). From 1991 to 2023, the density of hunted animals in Poland dropped sharply from approximately 850 to fewer than 200 individuals per 1000 km^2^ by the early 2000s and then stabilized at low levels [9]. In Lithuania, densities remained much lower, fluctuating modestly around 50–200 individuals per 1000 km^2^, and then stabilized as well. By the mid-2000s, both countries exhibited similarly low, steady hunting densities [28].

### 4.2. Hare Dietary Specialization and Habitat Requirements

In Central and Eastern Europe, *L. europaeus* demonstrates strong dietary selectivity for energy- and protein-rich forage. They depend on heterogeneous landscapes with permanent cover and diverse crops. Their populations are declining primarily due to agricultural intensification, which reduces dietary quality and habitat complexity. There is clear evidence linking agricultural practices, diet, and reproductive output [1,29,30,31]. Hares prefer plant material with a higher energy content, crude fat, and protein, while minimizing fiber. Selection for fat peaks in the summer. These traits are consistent across sex and age, but vary seasonally [32,33].

Hare diets shift seasonally to adapt to available resources. In the fall, they consume seeds, fruits, and legumes, though not necessarily in high proportions [34]. In the winter, when other foods may be scarce, *L. europaeus* eats woody species [35]. The rise in maize monocultures, which are low in nutritional value and niacin, leads to deficiencies that reduce the fecundity of female hares and the survival of their young, contributing to local population declines [31].

In Europe, *L. europaeus* exhibits a clear preference for fine-scale habitat features. They favor areas with short vegetation (1–25 cm), young cereal crops, field margins, and a mix of field structures. Permanent set-asides and fallows serve as crucial refuges for them [10,36]. They select sites with taller vegetation (>30 cm) for resting and refuge, particularly outside the main growing season. The loss of such cover is likely a key factor in declining resting-site quality and population decreases [37]. At the landscape scale, higher *L. europaeus* densities are associated with permanent grassland, winter cereals, oilseed rape, and greater crop-edge density. In contrast, large, uniform fields, expanding woodlands, and reduced edge habitat tend to lower densities [9,10,38,39].

These features enhance foraging opportunities and offer essential cover from predators and microhabitats that buffer against climatic extremes [40]. However, agricultural intensification, marked by landscape homogenization, monoculture expansion, and non-cropped habitat reduction, has significantly reduced food quality and habitat complexity. This has contributed to *L. europaeus* population declines across much of their range [29,30,41].

### 4.3. Agricultural Landscapes and Ecology of the European Brown Hare

The ecology and population dynamics of *L. europaeus* are strongly influenced by agricultural practices, with intensification being one of the most significant negative factors [41,42]. Brown hare populations have declined markedly as field sizes have increased and crop diversity has decreased, particularly with the rise in monocultures, such as maize. This decline is closely tied to the loss of habitat heterogeneity, which is essential for their survival and reproduction [30,31,43].

Agri-environmental programs have had limited success in supporting hare populations, often hindered by poor implementation [41,44]. In contrast, diverse cropping systems support higher hare densities compared to uniform landscapes lacking winter crops. Large, uniform patches without winter crops are particularly detrimental; conversely, more biodiverse and varied cropping systems tend to support higher hare densities [45].

Long-term data indicate an ongoing decline in European *L. europaeus* populations, primarily due to reduced juvenile recruitment in intensively farmed areas [1,2]. Permanent cover and non-grain crops increase densities, while grass fields, extreme temperatures, and monocultures have negative effects [9,10,30,42].

In summary, there is strong evidence that the nutritional and habitat needs of *L. europaeus* are closely linked to the quality of agricultural landscapes. However, studies linking individual diets to life-history traits in landscape contexts are scarce. Similarly, methods such as stable isotope analysis and integrated life-history assessments are underutilized.

### 4.4. Lepus Europaeus Diet Specific as Found from Isotopic Analysis

Through stable isotope analysis of *L. europaeus* hair, we revealed dietary distinctions in the samples from Lithuania and Poland, related to the sampled habitats. Lithuanian hares exhibited greater isotopic variability and a broader trophic niche, reflecting their diverse habitats and mixed diets of natural and cultivated plants. In contrast, Polish hares from orchard habitats exhibited higher δ^13^C values and narrower isotopic ranges, indicating a diet consisting primarily of cultivated vegetation.

Within Lithuania, the values of δ^13^C and δ^15^N varied by habitat. Riverine *L. europaeus* relied more on natural C_3_ plants, while hares in fields showed enrichment linked to fertilized crops. Seasonal patterns revealed higher δ^15^N values in winter, which is consistent with a shift toward woody or protein-rich foods. The effects of sex and age were not expressed.

Stable isotope analysis effectively links land-use intensity, habitat diversity, and trophic niche dynamics in farmland mammals, and our results demonstrate that the structure of landscapes influences the trophic ecology of *L. europaeus*. In Lithuania, sampled diverse habitats support broad dietary flexibility, whereas the narrower isotopic niche observed in Poland reflects that all sampled hares originated from a single, uniform orchard habitat, which provides more limited nutritional options compared with the multi-habitat sampling in Lithuania. This pattern should not be interpreted as representative of Poland as a whole, but rather as a consequence of the restricted habitat scope of our Polish sample.

Other than our own Polish *L. europaeus* δ^13^C data, the only available data from Selva et al. [24] show a trend of δ^13^C enrichment from 1950 to the 2000s. The earliest value, from 1950, is the most depleted at −29.49‰. After this, δ^13^C becomes progressively less negative over time. Values from the 1960s to the 1980s (−25.1 to −26.2‰) indicate gradual enrichment, followed by continued increases into the 2000s. By 2004–2009, δ^13^C ranges from −27.93 to −24.94‰, representing a shift of approximately +4.6‰ over ~60 years. However, comparing the long-term average with our data indicates that the two confidence intervals overlap, which is consistent with the non-significant *p*-value (t = 1.00, *p* = 0.32). The effect size is moderate (Cohen’s d = −0.48), suggesting that a real difference might exist, but the small *n* = 7 group in [24] limits power.

The δ^15^N data from [24] show substantial temporal variability rather than a consistent long-term directional trend, as the values ranged from −1.43‰ to 6.16‰, with several pronounced peaks (1963, 1987, and 2005) interspersed with depleted years (1950, 1986, 2004, and 2009). Instead of a steady increase or decrease, the δ^15^N record is characterized by episodic fluctuations. These shifts likely reflect short-term or localized changes in trophic position, diet composition, the availability of nitrogen-rich agricultural inputs, habitat use, movement among isotopically distinct areas, and variability in nitrogen cycling processes (e.g., soil moisture and microbial activity). Comparing the long-term average with our data indicates a nearly twofold increase (t = 1.99, *p* = 0.096) with a very large effect size (Cohen’s d = −1.03). Therefore, data from [24] are insufficient to draw a conclusion.

### 4.5. Ecological and Management Implications

Our findings confirm that the simplification of agricultural landscapes is a critical driver of the long-term decline of *L. europaeus* populations across Europe. Homogeneous, intensively managed farmlands reduce habitat heterogeneity and constrain dietary diversity, which ultimately limits the species’ capacity to adapt to seasonal and environmental fluctuations. These factors result in diminished resilience, lower reproductive success, and population instability [1,4].

Landscape heterogeneity, such as mixed cropping, grassy margins, scattered shrubs, and hedgerows, provides a broader trophic niche and structural refuge. This promotes higher hare densities and stable populations [4,32]. Long-term monitoring across Central and Eastern Europe has demonstrated that an increase in field size and monoculture dominance, particularly of maize, is negatively correlated with hare abundance and fecundity [2,31]. Conversely, the presence of semi-natural elements, permanent vegetation, and smaller fields supports population persistence [2,13].

Despite the introduction of agri-environmental schemes (AESs) intended to mitigate biodiversity loss, their effectiveness in protecting *L. europaeus* populations across Europe remains inconsistent. Many AES measures have failed due to poor targeting or insufficient promotion of structural diversity [4]. However, schemes that restore field margins, wildflower strips, and set-aside areas have increased food availability and cover, providing measurable benefits [2,4,32].

Studies on dietary adaptability highlight that brown hares are generalist herbivores capable of adjusting their diet according to the availability of local and seasonal vegetation [16]. However, agricultural intensification often leads to nutritional imbalances. For instance, large-scale maize cultivation can cause niacin deficiency in *L. europaeus*, which reduces reproductive output and impairs population development [31]. Similarly, reductions in plant species diversity and seasonal food shortages during summer harvests have been identified as critical limiting factors [13].

Furthermore, research integrating stable isotope analysis, stomach content analysis, and dental microwear reveals inconsistencies in dietary reconstructions, which complicates interpretations of feeding ecology in both extant and extinct lagomorphs [35]. These findings highlight the importance of using multiple ecological proxies to refine dietary and habitat models.

Overall, the interplay between agricultural intensification, forage quality, and reproductive ecology highlights the urgent need for landscape-level conservation strategies. Maintaining habitat heterogeneity through crop rotation, diversified farming systems, and restoring semi-natural vegetation are essential to reversing current declines. Long-term population data indicate that these measures can significantly increase recruitment and stabilize *L. europaeus* populations in intensively farmed regions [1,2]. These actions align with broader agroecological principles, emphasizing biodiversity-friendly management as a cornerstone for sustainable agroecosystems and the conservation of the European brown hare.

### 4.6. Study Limitations

One limitation of our study is the absence of δ^15^N and δ^13^C measurements from the plants in the sampled habitats. This leaves us without a vegetation baseline for interpreting the hare isotope values. C_3_ plants can exhibit isotopic variation across microhabitats, seasons, and plant parts, with differences in several per mil being driven by factors such as canopy cover, moisture, nitrogen sources, and plant physiology [46]. This variation is transferred directly into herbivore tissues. Though the fractionation of diets into hair in herbivores is relatively consistent (+2.7 to +3.5‰), the accurate interpretation of animal δ^13^C values necessitates an understanding of the isotopic composition of plants [47]. Without these data, it remains unclear whether the observed habitat-related differences reflect true dietary shifts or underlying vegetation isotopic variation. Therefore, future studies should include habitat-specific plant isotope sampling to distinguish dietary effects from baseline differences.

Another limitation is the unequal habitat composition and agricultural structure represented in our samples. Lithuanian hares were collected across multiple habitat types within a heterogeneous agricultural landscape. In contrast, all Polish hares originated from a single, structurally uniform orchard habitat. Consequently, differences attributed to “country” primarily reflect disparities in habitat diversity and land use rather than broader geographic patterns. This may limit the generalizability of our findings to Poland.

Although differences in age structure between the Lithuanian and Polish samples could be considered a limitation, our analyses showed no effect of age on δ^15^N or δ^13^C values. This indicates that age likely did not influence the observed country- or habitat-level patterns.

## 5. Conclusions

This study provides the first comprehensive, stable isotope-based assessment of the trophic ecology of the *Lepus europaeus* in Lithuania, as well as a comparative reference from Poland. Our systematic analysis of δ^13^C and δ^15^N values in hair revealed clear dietary differentiation between populations living in heterogeneous versus homogeneous agricultural landscapes. Namely, the habitat, rather than sex or age, explained most of the isotopic variation. This underscores the dominant influence of land-use structure on *L. europaeus* feeding ecology.

## Figures and Tables

**Figure 1 animals-16-00015-f001:**
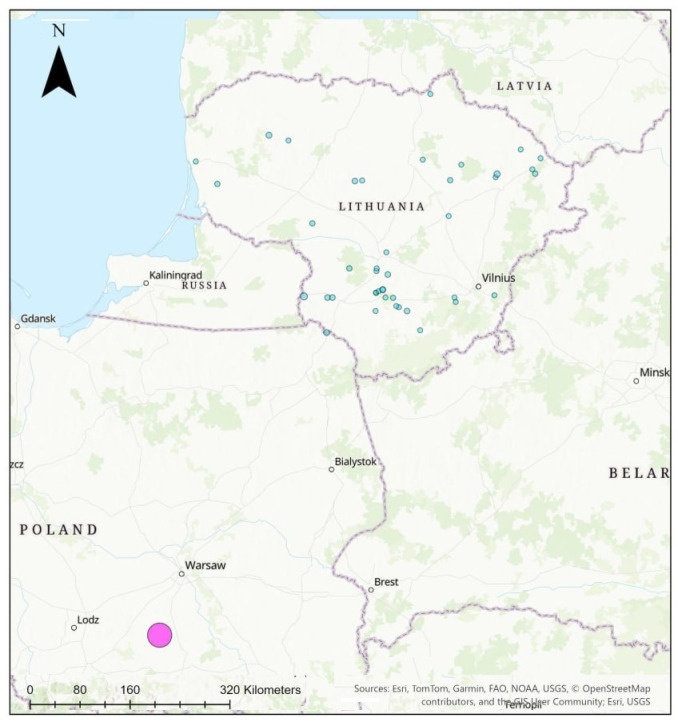
Sample collection sites of *L. europaeus* in Lithuania and Poland. Bubble size corresponds to sample size.

**Figure 2 animals-16-00015-f002:**
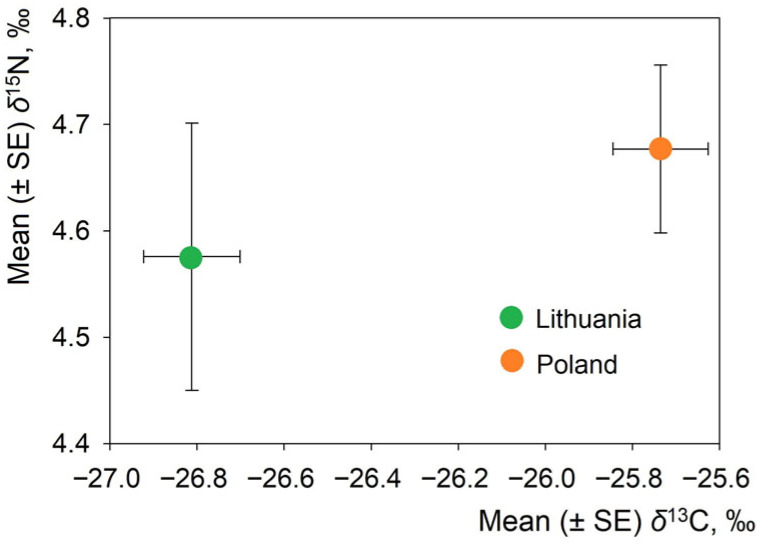
Position of *Lepus europaeus* in isotopic space according to stable isotope ratios in Lithuania and Poland.

**Figure 3 animals-16-00015-f003:**
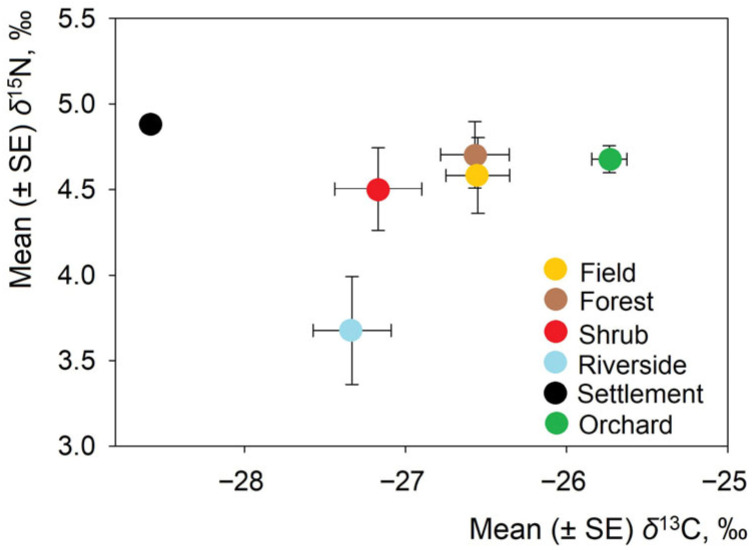
Position of *Lepus europaeus* in isotopic space according to stable isotope ratios in various habitats.

**Figure 4 animals-16-00015-f004:**
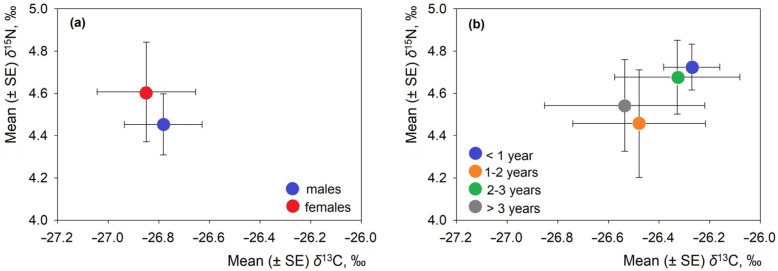
Position of *Lepus europaeus* males and females (**a**) and age groups (**b**) in isotopic space according to stable isotope ratios.

**Figure 5 animals-16-00015-f005:**
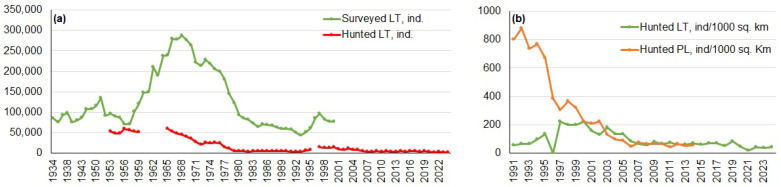
(**a**) Long-term trends in surveyed and hunted numbers of *Lepus europaeus* in Lithuania from 1934 to 2024. (**b**) Hunting bag density of *L. europaeus* in Lithuania and Poland from 1991 to 2024 (ind./1000 km^2^). Hunting bag densities were recalculated from the reported numbers of hunted individuals. Data sources: [9,28].

**Table 1 animals-16-00015-t001:** Sample size and breakdown by country, year, month, sex, and age class. Note: This information is not available for all individuals; therefore, the sums differ.

Country	Year			Month			Sex		Age Class			
	2023	2024	2025	Nov	Dec	Jan	M	F	I	II	III	IV
Lithuania	14	55	14	–	27	56	31	34	39	9	16	15
Poland		18	50	18	50	–	n/a	n/a	42	6	13	3

**Table 2 animals-16-00015-t002:** Central position (mean ± SD) and ranges of stable isotope ratios in the hair of *Lepus europaeus* in Lithuania and Poland.

Country	*n*	Mean δ^13^C (‰) ± SD	Range (Min–Max)	Mean δ^15^N (‰) ± SD	Range (Min–Max)
Lithuania	83	−26.81 ± 1.01	−28.61–−23.24	4.58 ± 1.14	2.30–9.80
Poland	68	−25.74 ± 0.90	−27.11–−20.60	4.68 ± 0.65	3.27–6.48

**Table 3 animals-16-00015-t003:** Central position (mean ± SD) and ranges of stable isotope ratios in the hair of *Lepus europaeus* from different habitats.

Habitat	*n*	Mean δ^13^C (‰) ± SD	Range (Min–Max)	Mean δ^15^N (‰) ± SD	Range (Min–Max)
Field	19	−26.55 ± 0.86	−27.92–−25.05	4.58 ± 0.96	3.25–6.22
Forest	27	−26.57 ± 1.11	−28.01–−23.24	4.70 ± 1.01	2.67–6.48
Shrub	17	−27.16 ± 1.11	−28.61–−24.73	4.50 ± 0.99	3.15–6.37
Riverside	6	−27.33 ± 0.59	−27.94–−26.47	3.68 ± 0.77	2.30–4.43
Settlement	1	−28.58	−28.58–−28.58	4.87	4.87–4.87
Orchard	68	−25.74 ± 0.90	−27.11–−20.60	4.68 ± 0.65	3.27–6.48

**Table 4 animals-16-00015-t004:** Temporal variation in central positions (mean ± SD) and ranges of stable isotope ratios in the hair of *Lepus europaeus*.

Country	Year/Month	*n*	Mean δ^13^C (‰) ± SD	Range (Min–Max)	Mean δ^15^N (‰) ± SD	Range (Min–Max)
Lithuania	2023	14	−26.92 ± 0.81	−28.01–−25.30	4.05 ± 0.86	2.67–6.01
	2024	55	−26.86 ± 1.12	−28.61–−23.24	4.75 ± 1.26	2.30–9.80
	2025	14	−26.53 ± 0.70	−27.36–−25.31	4.41 ± 0.67	3.42–5.67
Poland	2023	18	−26.36 ± 0.50	−27.11–−25.30	5.08 ± 0.43	3.96–5.60
	2024	50	−25.51 ± 0.91	−26.99–−20.60	4.53 ± 0.66	3.27–6.48
Lithuania	December	27	−26.69 ± 0.88	−28.01–−25.05	4.40 ± 0.98	2.67–6.48
	January	56	−26.87 ± 1.07	−28.61–−23.24	4.66 ± 1.21	2.30–9.80
Poland	November	18	−26.36 ± 0.50	−27.11–−25.30	5.08 ± 0.43	3.96–5.60
	December	50	−25.51 ± 0.91	−26.99–−20.60	4.53 ± 0.66	3.27–6.48

## Data Availability

We are ready to share raw data on the basis of scientific cooperation.

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
