# Peer review of "Animals2026, 16(1), 15;https://doi.org/10.3390/ani16010015"

_animals, 2025, doi:10.3390/ani16010015_

Round 1

Reviewer 1 Report

Comments and Suggestions for Authors

The study contains valuable information and is based on very fortunate selected methods. In my opinion several points have to be cleared. The study is very well written and I could prescribe minor revision, however I suggest rewriting of several paragraphs of the text. Because several changes in multiple paragraphs are needed, I have to decide on Major revision.

General remarks:

Please, use past tense everywhere it is possible!

Figure 1 is lacking resolution

In the results and in the discussion would be very interesting to be explained how particular isotope values measured in a particular habitat reflect diet shift. What are the relations between isotope values and plant composition?

“Lithuanian L. europaeus had significantly lower δ¹³C values (t = 6.83, p < 0.0001) than Polish individuals 274 (Table 2). However, δ¹⁵N values were similar between countries on average (t = 0.65, p = 275 0.52).” The authors have to discuss explicitly the material disbalance. The specimens from Lithuania were collected from wide variety of habitats and these from Poland were collected from one specific habitat. The age disbalance came from the fact, that the specimen from Poland were predominantly juveniles and subadults. This is important shortage of probes diversity, because in section 4.3 of the discussion the authors stated. “Through stable isotope analysis of L. europaeus hair, we revealed dietary, habitat, and geographic distinctions between Lithuania and Poland.” This has to be revised and the limitation of specimen collection and its implication have to be explicitly explained. The authors report on differences at State level and do not compare similar habitats in both EU Countries.

Figure 5 is unclear and has to be checked and explained

Please, explain the sentence from Discussion 4.3 “Lithuania’s diverse habitats support broad dietary flexibility, whereas Poland’s uniform orchards limit nutritional options.” There are for sure numerous suitable habitats for L. europaeus in Poland… The limitation of the collection of specimens only from orchards in Poland should be put in the focus of the paragraph.

The data presented in section 4.1 are rather confusing. Are these data based on existing literature sources, or were collected by the authors? Citations are needed in the text and not only in the figure captions.

The conclusion has to be revised. In the current form the second and the third paragraph contain information which was already provided. The authors may consider limitation of the section to the first paragraph.

Author Response

Rev #1 comments and answers

Comment: The study contains valuable information and is based on very fortunate selected methods. In my opinion several points have to be cleared. The study is very well written and I could prescribe minor revision, however I suggest rewriting of several paragraphs of the text. Because several changes in multiple paragraphs are needed, I have to decide on Major revision.

Answer: thank you for your comments.

General remarks:

Comment: Please, use past tense everywhere it is possible!

Answer: thank you, we went through the text using the rule of thumb: Past tense when done in the past (methods, results, previous studies), and the Present tense if statement true now or always true.

No changes of tense are needed in the Abstract or Introduction. The Material and Methods section is now in the past tense. The Results section is in the past tense. The Discussion section uses both the present and past tenses, depending on the context.

Comment: Figure 1 is lacking resolution

Answer: uploaded file had 300x300 resolution, now we cropped part of it, so in published form it will be detailed enough.

Comment: In the results and in the discussion would be very interesting to be explained how particular isotope values measured in a particular habitat reflect diet shift. What are the relations between isotope values and plant composition?

Answer: Limitation of our study – we did not sample or analyse carbon and nitrogen isotope values of plants in the habitats where hares were collected. Consequently, we lack ecosystem-specific isotopic baselines against which hare hair δ¹³C and δ¹⁵N values could be directly interpreted. Research on C₃ plant systems shows that δ¹³C and δ¹⁵N values vary strongly within and among habitats, even when only C₃ vegetation is present. Metcalfe (2021) demonstrated that C₃ plant isotope values differ significantly across microhabitats (open, closed, wet) and plant parts, with shifts of several per mil occurring due to canopy cover, water availability, nitrogen sources, and seasonal changes These variations can pass directly into herbivore tissues, meaning that animals feeding on the same plant species in different microhabitats may exhibit different isotope values even without altering the botanical composition of their diet.

Similarly, controlled-feeding studies show that carbon isotope discrimination between diet and hair is relatively consistent (+2.7 to +3.5‰), but accurate dietary reconstruction requires knowing the δ¹³C values of the consumed plants themselves (Sponheimer et al. 2003). Because we did not quantify δ¹³C of plant communities, we cannot determine whether, for example, higher δ¹³C values in orchard habitats reflect increased consumption of specific crops or simply that plants grown in orchard microhabitats have intrinsically higher δ¹³C values due to environmental stress, management practices, or canopy structure.

Therefore, although our results demonstrate clear isotopic differentiation among habitats, we cannot attribute these patterns unambiguously to dietary shifts in plant species composition. Instead, the variations may partially (or substantially) reflect baseline isotopic heterogeneity of C₃ vegetation across Lithuanian agricultural mosaics and Polish orchards. Future research should incorporate systematic sampling of local plants, following the recommendations of Metcalfe (2021), to disentangle plant-driven baseline variation from true trophic shifts and to improve the precision of dietary interpretations based on stable isotopes.

We added this as Study limitations. Nevertheless, δ¹³C values still indicate hare diet being based on C₃ vegetation, as was shown in the manuscript.

Comment: “Lithuanian L. europaeus had significantly lower δ¹³C values (t = 6.83, p < 0.0001) than Polish individuals 274 (Table 2). However, δ¹⁵N values were similar between countries on average (t = 0.65, p = 275 0.52).” The authors have to discuss explicitly the material disbalance. The specimens from Lithuania were collected from wide variety of habitats and these from Poland were collected from one specific habitat. The age disbalance came from the fact, that the specimen from Poland were predominantly juveniles and subadults. This is important shortage of probes diversity, because in section 4.3 of the discussion the authors stated. “Through stable isotope analysis of L. europaeus hair, we revealed dietary, habitat, and geographic distinctions between Lithuania and Poland.” This has to be revised and the limitation of specimen collection and its implication have to be explicitly explained. The authors report on differences at State level and do not compare similar habitats in both EU Countries.

Answer: great comment, we used country names by default, so a few revisions were really needed.

First, we inserted explanation into the last paragraph of Introducion, cole to the Aim:” Because the country identity in our study reflects the sampling design rather than true nationwide variation, with Lithuania encompassing several habitat types and Poland comprising only a single orchard site, the observed differences largely reflect habitat scope rather than geopolitical boundaries.”

Second, we re-worded beginning of 4.3., as “Through stable isotope analysis of L. europaeus hair, we revealed dietary distinctions in the samples from Lithuania and Poland, related to sampled habitats.”

Finally, unevenness of sampling we also acknowledge in 4.5 section as study limitation.

Comment: Figure 5 is unclear and has to be checked and explained

Answer: Figure 5 is correct. We propose different caption for clearness. Figure 5. (a) Long-term trends in surveyed and hunted numbers of Lepus europaeus in Lithuania from 1934 to 2024. (b) Hunting bag density of L. europaeus in Lithuania and Poland from 1991 to 2024 (ind./1000 km²). Data sources: [28,9].

Comment: Please, explain the sentence from Discussion 4.3 “Lithuania’s diverse habitats support broad dietary flexibility, whereas Poland’s uniform orchards limit nutritional options.” There are for sure numerous suitable habitats for L. europaeus in Poland… The limitation of the collection of specimens only from orchards in Poland should be put in the focus of the paragraph.

Answer: We thank for this comment, as the wording was really misleading. Of course, Poland provides numerous suitable habitats for L. europaeus, but our statement was not intended to generalize the ecological conditions of the entire country. The observed pattern reflects only the specific sampling context of our study, in which all Polish individuals originated from a single, uniform orchard landscape. We have revised the text to clarify that the narrower dietary niche observed in Poland reflects the restricted habitat range of the collected samples, not a national-level habitat limitation. This clarification has now been added to the Discussion:

Stable isotope analysis effectively links land-use intensity, habitat diversity, and trophic niche dynamics in farmland mammals, and our results demonstrate that the structure of landscapes influences the trophic ecology of L. europaeus. In Lithuania, sampled diverse habitats support broad dietary flexibility, whereas the narrower isotopic niche observed in Poland reflects that all sampled hares originated from a single, uniform orchard habitat, which provides more limited nutritional options compared with the multi-habitat sampling in Lithuania. This pattern should not be interpreted as representative of Poland as a whole, but rather as a consequence of the restricted habitat scope of our Polish sample.

Comment: The data presented in section 4.1 are rather confusing. Are these data based on existing literature sources, or were collected by the authors? Citations are needed in the text and not only in the figure captions.

Answer: The data presented in Section 4.1 indeed originate from existing literature and archival sources, not from our own field collection. To improve clarity, we have added explicit in-text citations to the population and hunting statistics referenced in this section. Lithuanian long-term survey data, and hunting data in both countries are taken from national game statistics [28] or the published paper [9]. These citations now appear directly in the text, not only in the figure caption, ensuring that the provenance of the data is transparent.

Comment: The conclusion has to be revised. In the current form the second and the third paragraph contain information which was already provided. The authors may consider limitation of the section to the first paragraph.

Answer: We deleted the third paragraph, and reworded the second to merge with the first.

Reviewer 2 Report

Comments and Suggestions for Authors Carbon and nitrogen hair isotopes analysis was used to assess the trophic ecology of European Brown hares (EBh) by examining their dietary niches across different habitats, sexes, and age groups. The results demonstrate that habitat heterogeneity is a primary factor influencing dietary flexibility in L. europaeus. Since habitat quality, rather than sex or age, is closely associated with agricultural land use, these findings are significant for agroecosystem management and the conservation of brown hares. My opinion of this manuscript is very favorable; I will only point out a few details that I believe will contribute to improving the current presentation of this study. While the title is not inappropriate, the term "exploratory" is usually used for smaller datasets, less detailed analyses, or less conclusive results. In this case, a more informative title that better represents the contribution of this study will be more useful to readers. The introduction is clear and provides the most relevant background information to establish the theoretical framework and context needed to understand the objectives of the study. Particularly, changes in population density of EBh in Europe, the main factors attributed to mortality, and the relationship between habitat characteristics and population decline are examined. Also, changes in the diet of EBh according to different environments, including those under agricultural practices. Concomitantly, alternative methods to study diet in mammals is presented, including the one used here, carbon and nitrogen isotopes. In the specific context of this study, more basic information about the information provided by each isotope is required. The main objective explains that the diet of hares inhabiting different environments in Lithuania will be analyzed to determine the species' trophic niche in relation to environmental changes. A comparison with results from Poland (orchard ambient) is also proposed. The identification of countries is not relevant here; rather, the characterization of the environments being compared is. It would be advisable for the objective to be more informative regarding the environments considered in the comparison. Furthermore, the presentation of hypotheses and predictions is suggested. Materials and methods are clear and show the necessary details. According to the above commentary, “country” is not a variable here; instead, ambient must be used and “Poland” changed by “orchard”. Results. The data obtained are properly analyzed, and the results are clearly reported in the text, tables, and graphs.
The discussion section is very well structured. First, population fluctuations of hares over time are shown for Lithuania and Poland, along with dietary preferences and their relationship to habitat characteristics, particularly the influence of agricultural practices. Then, hair isotope analysis revealed that environmental variability in Lithuania corresponds to greater isotopic variability and a broader trophic niche. In Poland, the orchard environment, with cultivated vegetation, is associated with higher carbon isotope values ​​and a narrower range. Changes in nitrogen isotopes are discussed in terms of their temporal variability, indicating possible sources for this variation. This study shows that isotope evaluation reveals differences in the diet of hares associated with environmental heterogeneity.

Author Response

Rev #2 comments and answers

Comment: Carbon and nitrogen hair isotopes analysis was used to assess the trophic ecology of European Brown hares (EBh) by examining their dietary niches across different habitats, sexes, and age groups. The results demonstrate that habitat heterogeneity is a primary factor influencing dietary flexibility in L. europaeus. Since habitat quality, rather than sex or age, is closely associated with agricultural land use, these findings are significant for agroecosystem management and the conservation of brown hares. My opinion of this manuscript is very favorable; I will only point out a few details that I believe will contribute to improving the current presentation of this study.

Answer: thank you, below we present all answers to your comments and description of revisions.

Comment: While the title is not inappropriate, the term "exploratory" is usually used for smaller datasets, less detailed analyses, or less conclusive results. In this case, a more informative title that better represents the contribution of this study will be more useful to readers.

Answer: the new title is “Stable Isotope Analysis Reveals Habitat-Driven Dietary Niches of Lepus europaeus”.

Comment: The introduction is clear and provides the most relevant background information to establish the theoretical framework and context needed to understand the objectives of the study. Particularly, changes in population density of EBh in Europe, the main factors attributed to mortality, and the relationship between habitat characteristics and population decline are examined. Also, changes in the diet of EBh according to different environments, including those under agricultural practices. Concomitantly, alternative methods to study diet in mammals is presented, including the one used here, carbon and nitrogen isotopes.

Answer: therefore, no changes requested in revision.

Comment: In the specific context of this study, more basic information about the information provided by each isotope is required.

Answer: We suppose, that most of the information is already in the manuscript. But still, to acknowledge your comment, we added additional paragraph in 1.3.: “Thus, carbon (δ¹³C) and nitrogen (δ¹⁵N) isotopes provide complementary dietary information in herbivores. δ¹³C reflects the types of plants assimilated and differences among habitats or agricultural resources. Meanwhile, δ¹⁵N indicates trophic position and protein sources and often increases with the consumption of nitrogen-rich or fertilized plants. Since keratin integrates assimilated nutrients over a period of weeks to months, stable isotope ratios in hair provide an averaged representation of diet and habitat use over time [18–20].”

Comment: The main objective explains that the diet of hares inhabiting different environments in Lithuania will be analyzed to determine the species' trophic niche in relation to environmental changes. A comparison with results from Poland (orchard ambient) is also proposed. The identification of countries is not relevant here; rather, the characterization of the environments being compared is. It would be advisable for the objective to be more informative regarding the environments considered in the comparison.

Answer: We thank the reviewer for this observation. We agree that we should emphasize the environments represented in our sample rather than the countries' geopolitical identities. However, our study does not aim to evaluate environmental change over time. Instead, it focuses on comparing hare diets across the different habitat types included in our Lithuanian sample and the single orchard habitat represented in the Polish sample. As clarified in the revised manuscript, "country" strictly functions as a sampling descriptor because Lithuania encompasses multiple habitat types, whereas Poland is represented by a single orchard site. Accordingly, we revised the objective to more explicitly describe the environments included in the comparison, avoiding any implication that national-level or temporal environmental changes were being assessed. The revised wording accurately reflects that the study examines habitat-driven dietary differences rather than environmental change.

We added explanation to the Aim, that “By analysing individuals from contrasting environments, we provide a contemporary baseline for the species’ isotopic niche within modern agricultural landscapes.”

Comment: Furthermore, the presentation of hypotheses and predictions is suggested. Materials and methods are clear and show the necessary details. According to the above commentary, “country” is not a variable here; instead, ambient must be used and “Poland” changed by “orchard”.

Answer: We appreciate the suggestions to include explicit hypotheses and replace "country" with "ambient." In our view, the study's aims are clearly stated and the analytical framework is straightforward. Thus, adding formal hypotheses would not substantially improve clarity. However, we have incorporated the reviewer’s recommendation by clarifying throughout the manuscript that "country" does not represent a biological or geopolitical variable but rather reflects the sampling context: multiple habitat types in Lithuania versus a single orchard environment in Poland. Accordingly, we now explicitly refer to the Polish material as orchard habitat rather than a country-level comparison in all relevant sections. With these revisions, we believe the purpose and structure of the study are clear.

Comment: Results. The data obtained are properly analyzed, and the results are clearly reported in the text, tables, and graphs.

Answer: thank you

Comment: The discussion section is very well structured. First, population fluctuations of hares over time are shown for Lithuania and Poland, along with dietary preferences and their relationship to habitat characteristics, particularly the influence of agricultural practices. Then, hair isotope analysis revealed that environmental variability in Lithuania corresponds to greater isotopic variability and a broader trophic niche. In Poland, the orchard environment, with cultivated vegetation, is associated with higher carbon isotope values ​​and a narrower range. Changes in nitrogen isotopes are discussed in terms of their temporal variability, indicating possible sources for this variation. This study shows that isotope evaluation reveals differences in the diet of hares associated with environmental heterogeneity.

Answer: We thank the reviewer for the positive evaluation of our discussion and for accurately summarizing the main points linking hare dietary variation to environmental heterogeneity. Our findings indeed show that structurally diverse Lithuanian landscapes correspond to broader isotopic variability and wider trophic niches. In contrast, the more uniform orchard system in Poland is associated with higher δ¹³C values and a narrower isotopic range. Similarly, we presume that the observed temporal variability in δ¹⁵N likely reflects environmental or physiological changes rather than significant shifts in trophic level.

However, we must emphasize an important study limitation (included in Revision 1): we did not measure δ¹⁵N or δ¹³C values in plants from the sampled habitats. Therefore, we lack a vegetation baseline against which hare isotope values can be directly interpreted. As ecosystem-scale studies have shown (e.g., Metcalfe, 2021), C₃ plants can vary isotopically across microhabitats and seasons. Controlled-feeding studies have demonstrated that accurately interpreting hair δ¹³C requires knowledge of plant isotopic composition (Sponheimer et al., 2003). Thus, although our results clearly illustrate habitat-associated isotopic differences in hares, some of this variation may reflect underlying differences in plant isotope values rather than diet composition alone. Incorporating plant baseline sampling in future work would allow for a more precise separation of dietary shifts from environmental isotopic variability.

Round 2

Reviewer 1 Report

Comments and Suggestions for Authors

The revised version of the manuscript can be accepted for publication.